# Saccader: Improving Accuracy of Hard Attention Models for Vision

**Gamaleldin F. Elsayed**
Google Research, Brain Team
gamaleldin@google.com

**Simon Kornblith**
Google Research, Brain Team

**Quoc V. Le**
Google Research, Brain Team

## Abstract

Although deep convolutional neural networks achieve state-of-the-art performance across nearly all image classification tasks, their decisions are difficult to interpret. One approach that offers some level of interpretability by design is *hard attention*, which uses only relevant portions of the image. However, training hard attention models with only class label supervision is challenging, and hard attention has proved difficult to scale to complex datasets. Here, we propose a novel hard attention model, which we term Saccader. Key to Saccader is a pretraining step that requires only class labels and provides initial attention locations for policy gradient optimization. Our best models narrow the gap to common ImageNet baselines, achieving 75% top-1 and 91% top-5 while attending to less than one-third of the image.

## 1 Introduction

Despite the success of convolutional neural networks (CNNs) across many computer vision tasks, their predictions are difficult to interpret. Because CNNs compute complex nonlinear functions of their inputs, it is often unclear what aspects of the input contributed to the prediction. Although many researchers have attempted to design methods to interpret predictions of off-the-shelf CNNs [Zhou et al., 2016, Baehrens et al., 2010, Simonyan and Zisserman, 2014, Zeiler and Fergus, 2014, Springenberg et al., 2014, Bach et al., 2015, Yosinski et al., 2015, Nguyen et al., 2016, Montavon et al., 2017, Zintgraf et al., 2017], it is unclear whether these explanations faithfully describe the underlying model [Kindermans et al., 2017, Adebayo et al., 2018, Hooker et al., 2018, Rudin, 2019]. Additionally, adversarial machine learning research [Szegedy et al., 2013, Goodfellow et al., 2017, 2014] has demonstrated that imperceptible modifications to inputs can change classifier predictions, underscoring the unintuitive nature of CNN-based image classifiers.

One interesting class of models that offers more interpretable decisions are "hard" visual attention models. These models rely on a controller that selects relevant parts of the input to contribute to the decision, which provides interpretability by design. These models are inspired by human vision, where the fovea and visual system process only a limited portion of the visual scene at high resolution [Wandell, 1995], and top-down pathways control eye movements to sequentially sample salient parts of visual scenes [Schütz et al., 2011]. Although models with hard attention perform well on simple datasets [Larochelle and Hinton, 2010, Mnih et al., 2014, Ba et al., 2014, Gregor et al., 2015], it has been challenging to scale these models from small tasks to real world images [Sermanet et al., 2015].

Here, we propose a novel hard visual attention model that we name Saccader, as well as an effective procedure to train this model. The Saccader model learns features for different patches in the image that reflect the degree of relevance to the classification task, then proposes a sequence of image patches for classification. Our pretraining procedure overcomes the sparse-reward problem that makes hard attention models difficult to optimize. It requires access to only class labels and provides initial attention locations. These initial locations provide better rewards for the policy gradient learning. Our

**(a)**

Saccader (learned policy)

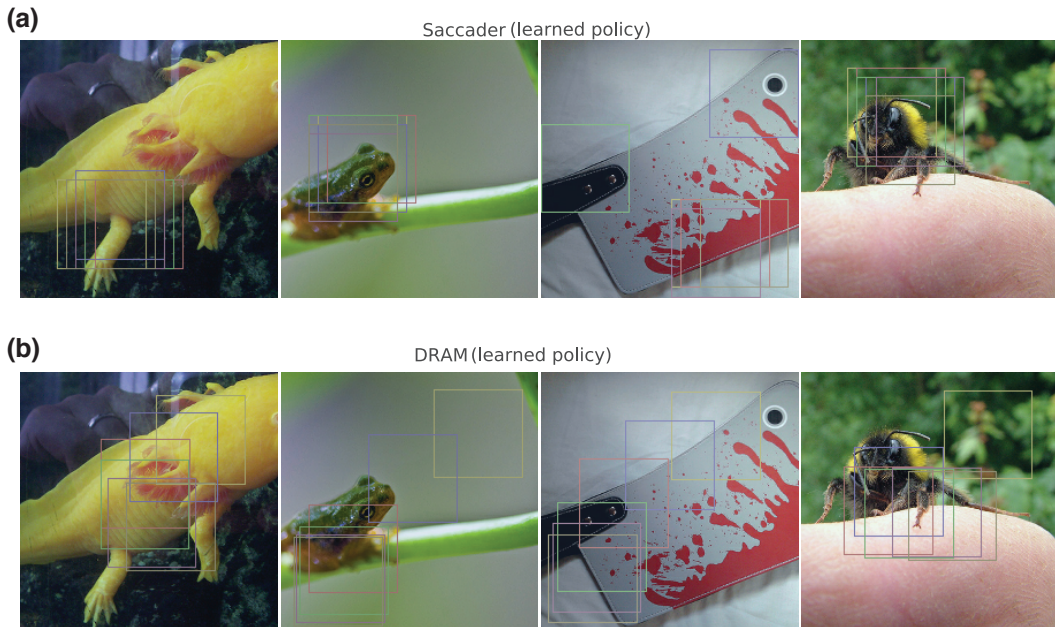

**(b)**

DRAM (learned policy)

Figure 1: **Examples of visual attention policies. (a)** Glimpses predicted by the Saccader model (more examples are shown in Figure Supp.1). **(b)** Glimpses predicted by the DRAM model [Ba et al., 2014, Sermanet et al., 2015].

results show that the Saccader model is highly accurate compared to other visual attention models while remaining interpretable (Figure 1). Our best models narrow the gap to common ImageNet baselines, achieving 75% top-1 and 91% top-5 while attending to less than one-third of the image. We further demonstrate that occluding the image patches proposed by the Saccader model highly impairs classification, thus confirming these patches strong relevance to the classification task.

## 2 Related Work

**Hard attention:**  Models that employ *hard attention* make decisions based on only a subset of pixels in the input image, typically in the form of a series of glimpses. These models typically utilize some sort of artificial fovea controlled by an adaptive controller that selects parts of the input to be processed [Burt, 1988, Ballard et al., 1988, Ballard, 1989, Seibert and Waxman, 1989, Ahmad, 1992, Olshausen et al., 1993]. Early work using backpropagation to train the controller computed the gradient with respect to its weights by backpropagating through a separate neural network model of the environmental dynamics [Schmidhuber and Huber, 1990, 1991b,a]. Butko and Movellan [2008] proposed instead to use policy gradient, learning a convolutional logistic policy to maximize long-term information gain. Later work extended the policy gradient framework to policies parameterized by neural networks to perform image classification [Mnih et al., 2014, Ba et al., 2014] and image captioning [Xu et al., 2015]. Other non-policy-gradient-based approaches include direct estimation of the probability of correct classification for each glimpse location [Larochelle and Hinton, 2010, Zheng et al., 2015] or differentiable attention based on adaptive downsampling [Gregor et al., 2015, Jaderberg et al., 2015, Eslami et al., 2016].

Work employing hard attention for vision tasks has generally examined performance only on relatively simple datasets such as MNIST and SVHN [Mnih et al., 2014, Ba et al., 2014]. Sermanet et al. [2015] adapted the model proposed by Ba et al. [2014] to classify the more challenging Stanford Dogs dataset, but the model achieved only a modest accuracy improvement over a non-attentive baseline, and did not appear to derive significant benefit from glimpses beyond the first.

**Soft attention:**  Models with hard attention are difficult to train with gradient-based optimization. To make training more tractable, other models have resorted to *soft attention* [Bahdanau et al., 2014,

[Xu et al., 2015]. Typical soft attention mechanisms rescale features at one or more stages of the network. The soft masks used for rescaling often appear to provide some insight into the model's decision-making process [Xu et al., 2015, Das et al., 2017], but the model's final decision may nonetheless rely on information provided by features with small weights [Jain and Wallace, 2019].

Soft attention is popular in models used for natural language tasks [Bahdanau et al., 2014, Luong et al., 2015, Vaswani et al., 2017], image captioning [Xu et al., 2015, You et al., 2016, Rennie et al., 2017, Lu et al., 2017, Chen et al., 2017], and visual question answering [Andreas et al., 2016], but less common in image classification. Although several spatial soft attention mechanisms for image classification have been proposed [Wang et al., 2017, Jetley et al., 2018, Woo et al., 2018, Linsley et al., 2019, Fukui et al., 2019], current state-of-the-art models do not use these mechanisms [Zoph et al., 2018, Liu et al., 2018a, Real et al., 2018, Huang et al., 2018]. The squeeze-and-excitation block, which was a critical component of the model that won the 2017 ImageNet Challenge [Hu et al., 2018], can be viewed as a form of soft attention, but operates over feature channels rather than spatial dimensions.

**Supervised approaches:** Although our aim in this work is to perform classification with only image-level class labels, our approach bears some resemblance to two-stage object detection models. These models operate by generating many region proposals and then applying a classification model to each proposal [Uijlings et al., 2013, Girshick et al., 2014, Girshick, 2015, Ren et al., 2015]. Unlike our work, these approaches use ground-truth bounding boxes to train the classification model, and modern architectures also use bounding boxes to supervise the proposal generator [Ren et al., 2015].

# 3 Methods

## 3.1 Architecture

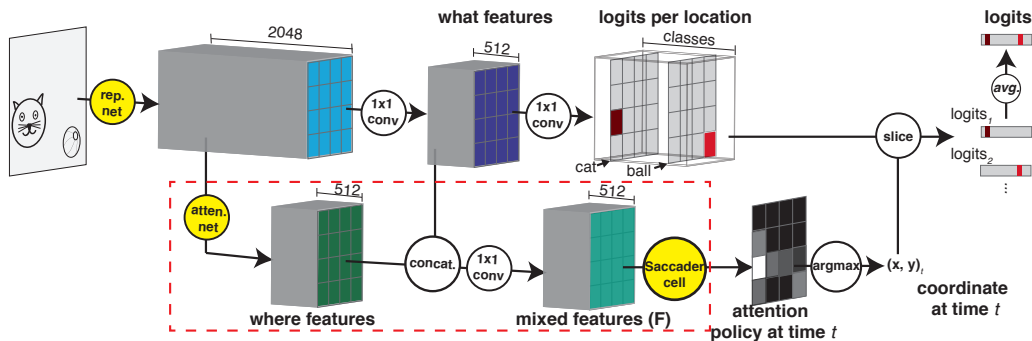

Figure 2: **Model architecture.** (a) Illustration of the Saccader model. The proposed location network pretraining is performed on components within the dashed red box. Components in yellow are described in detail in Figures Supp.4, Supp.5, and Supp.6

To understand the intuition behind the Saccader model architecture (Figure 2), imagine one uses a trained ImageNet model and applies it at different locations of an image to obtain logits vectors at these locations. To find the correct label, one could compute a global average of these vectors, and to find a salient location on the image, one reasonable choice is the location of the patch the elicits the largest response. With that intuition in mind, we design the Saccader model to compute sets of 2D features for different patches in the image, and select one of these sets of features at each time (Figure Supp.6). The attention mechanism learns to select the most salient location at the current time. For each predicted location, one may extract the logits vector and perform averaging across different times to perform class prediction.

In particular, our architecture consists of the following three components[1]:

1. Representation network: This is a CNN that processes glimpses from different locations of an image (Figure 2). To restrict the size of the receptive field (RF), one could divide the image into

patches and process each separately with an ordinary CNN, but this is computationally expensive. Here, we used the "BagNet" architecture from Brendel and Bethge [2019], which enables us to compute representations with restricted RFs efficiently in a single pass, without scanning. Similar to Brendel and Bethge [2019], we use a ResNet architecture where most $3 \times 3$ convolutions are replaced with $1 \times 1$ convolutions to limit the RF of the model and strides are adjusted to obtain a higher resolution output (Figure Supp.4). Our model has a $77 \times 77$ pixel RF and computes 2048-dimensional feature vectors at different locations in the image separated by only 8 pixels. For $224 \times 224$ pixel images, this maps to 361 possible attention locations. Before computing the logits, we apply a $1 \times 1$ convolution with ReLU activation to encode the representations into a 512-dimensional feature space ("what" features; name motivated by the visual system ventral pathway [Goodale and Milner, 1992]), and then apply another $1 \times 1$ convolution to produce the 1000-dimensional logits tensor for classification. We find that introducing this bottleneck provides a small performance improvement over the original BagNet model; we term the modified network BagNet-lowD.

2. Attention network: This is a CNN that operates on the 2048-dimensional feature vectors (Figure 2), and includes 4 convolutional layers alternating between $1 \times 1$ convolution and $3 \times 3$ dilated convolution with rate 2, each followed by batch normalization and ReLU activation (see Figure Supp.5). The $1 \times 1$ convolution layers reduce the dimensionality from 2048 to 1024 to 512 location features while the $3 \times 3$ convolutional layers widen the RF ("where" features; name motivated by the visual system dorsal pathway [Goodale and Milner, 1992]). The what and where features are then concatenated and mixed using a linear $1 \times 1$ convolution to produce a compact tensor with 512 features ($F$).

3. Saccader cell: This cell takes the mixed what and where features $F$ and produces a sequence of location predictions. Elements in the sequence correspond to target locations (Figure Supp.6). The cell includes a 2D state ($C^t$) that keeps memory of the visited locations until time $t$ by placing 1 in the corresponding location in the cell state. We use this state to prevent the network from returning to previously seen locations. The cell first selects relevant spatial locations from $F$ and then selects feature channels based on the relevant locations:

$$G_{ij}^t = \sum_{p=1}^{d} \frac{F_{ijp} a_p}{\sqrt{d}} - 10^5 C_{ij}^{t-1} \qquad \tilde{G}_{ij}^t = \frac{\exp(G_{ij}^t)}{\sum_{m=1}^{h} \sum_{n=1}^{w} \exp(G_{mn}^t)} \qquad (1)$$

$$h_k^t = \sum_{i=1}^{h} \sum_{j=1}^{w} F_{ijk} \tilde{G}_{ij}^t \qquad \tilde{h}_k^t = \frac{\exp(h_k^t)}{\sum_{p=1}^{d} \exp(h_p^t)} \qquad (2)$$

where $h$ and $w$ are the height and width of the output features from the representation network, $d$ is the dimensionality of the mixed features, and $a \in \mathbf{R}^d$ is a trainable vector. We use a large negative number multiplied by the state (i.e., $-10^5 C_{ij}^{t-1}$) to mask out previously used locations. Next, the cell computes a weighted sum of the feature channels and performs a spatial softmax to compute the policy:

$$R_{ij}^t = \sum_{k=1}^{d} F_{ijk} \tilde{h}_k^{\mathrm{t}} - 10^5 C_{ij}^{t-1} \qquad \tilde{R}_{ij}^t = \frac{\exp(R_{ij}^t)}{\sum_{m=1}^{h} \sum_{n=1}^{w} \exp(R_{mn}^t)} \qquad (3)$$

$\tilde{R}$ reflects the model's policy over glimpse locations. At test time, the model extracts the logits at time $t$ from the representation network at location $\arg\max_{i,j}(\tilde{R}_{ij}^t)$. The final prediction is obtained by averaging the extracted logits across all times.

In terms of complexity, the Saccader model has 35,583,913 parameters, which is 20% fewer than the 45,610,219 parameters in the DRAM model (Table Supp.1).

## 3.2 Training Procedure

In all our training, we divide the standard ImageNet ILSVRC 2012 training set into training and development subsets. We trained our model on the training subset and chose our hyperparameters based on the development subset. We follow common practice and report results on the separate ILSVRC 2012 validation set, which we do not use for training or hyperparameter selection. The goal

of our training procedure is to learn a policy that predicts a sequence of visual attention locations that is useful to the downstream task (here image classification) in absence of location labels.

We performed a three step training procedure using only the training class labels as supervision. First, we pretrained the representation network by optimizing the cross entropy loss computed based on the average logits across all possible locations plus $\ell^2$-regularization on the model weights. More formally, we optimize:

$$J(\theta) = -\log\left(\frac{\prod_{i=1}^{h}\prod_{j=1}^{w}P_\theta(y_{\text{target}}|X^{ij})^{\frac{1}{hw}}}{\sum_{k=1}^{c}\prod_{i=1}^{h}\prod_{j=1}^{w}P_\theta(y_k|X^{ij})^{\frac{1}{hw}}}\right) + \frac{\lambda}{2}\sum_{i=1}^{N}\theta_i^2 \quad (4)$$

where $X^{ij} \in \mathbf{R}^{77\times77\times3}$ is the image patch at location $(i,j)$, $y_{\text{target}}$ is the target class, $c = 1000$ is the number of classes, $\theta$ are the representation network parameters, and $\lambda$ is a hyperparameter.

Second, we use self-supervision to pretrained the location network (i.e., attention network, $1 \times 1$ mixing convolution and Saccader cell) to emit glimpse locations ordered by descending value of the logits. Just as SGD biases neural network training toward solutions that generalize well, the purpose of this pretraining is to alter the training trajectory in a way that produces a better-performing model. Namely, we optimized the following objective:

$$J(\eta) = -\log\left(\prod_{t=1}^{T}\pi_{\theta,\eta}(l_{\text{target}}^{t}|X, C^{t-1})\right) + \frac{\nu}{2}\sum_{i=1}^{N}\eta_i^2 \quad (5)$$

where $l_{\text{target}}^{t}$ is the $t^{\text{th}}$ sorted target location, i.e., $l_{\text{target}}^{1}$ is the location with largest maximum logit, and $l_{\text{target}}^{361}$ is the location with the smallest maximum logit. $\pi_{\theta,\eta}(l_{\text{target}}^{t}|X, C^{t-1})$ is the probability the model gives for attending to location $l_{\text{target}}^{t}$ at time $t$ given the input image $X \in \mathbf{R}^{224\times224\times3}$ and cell state $C^{t-1}$, i.e. $\tilde{R}_{ij}^{t}$ where $(i,j) = l_{\text{target}}^{t}$. The parameters $\eta$ are the weights of the attention network and Saccader cell. For this step, we fixed $T = 12$.

Finally, we trained the whole model to maximize the expected reward, where the reward ($r \in \{0,1\}$) represents whether the model final prediction after 6 glimpses ($T = 6$) is correct. In particular, we used the REINFORCE loss [Williams, 1992] for discrete policies, cross entropy loss and $\ell^2$-regularization. The parameter update is given by the gradient of the objective:

$$J(\theta,\eta) = -\sum_{s=1}^{S}\left(\log\left(\prod_{t=1}^{T}\pi_{\theta,\eta}(l_s^t|X, C_s^{t-1})\right)\right)(r_s - b) + \frac{\nu}{2}\sum_{i=1}^{N}\eta_i^2 \quad (6)$$
$$-\log\left(\frac{\prod_{t=1}^{T}P_\theta(y_{\text{target}}|X^t)^{1/T}}{\sum_{k=1}^{c}\prod_{t=1}^{T}P_\theta(y_k|X^t)^{1/T}}\right) + \frac{\lambda}{2}\sum_{i=1}^{N}\theta_i^2$$

where we sampled $S = 2$ trajectories $l_s$ at each time from a categorical distribution with location probabilities given by $\pi_{\theta,\eta}(l|X, C_s^{t-1})$, $b$ is the average accuracy of the model computed on each minibatch, and $X^t$ denotes the image patch sampled at time $t$. The role of adding $b$ and the $S$ Monte Carlo samples is to reduce variance in our gradient estimates [Sutton et al., 2000, Mnih et al., 2014].

In each of the above steps, we trained our model for 120 epochs using Nesterov momentum of 0.9. (See Appendix for training hyperparameters).

## 4 Results

In this section, we use the Saccader model to classify the ImageNet (ILSVRC 2012) dataset. ImageNet is an extremely large and diverse dataset that contains both coarse- and fine-grained class distinction. To achieve high accuracy, a model must not only distinguish among superclasses, but also e.g. among the $> 100$ fine-grained classes of dogs. We show that the Saccader model learns a policy that yields high accuracy on ImageNet classification task compared to other learned and engineered policies. Moreover, we show that our pretraining for the location network helps achieve that high accuracy. Finally, we demonstrate that the attention locations proposed by our model are highly relevant to the classification task.

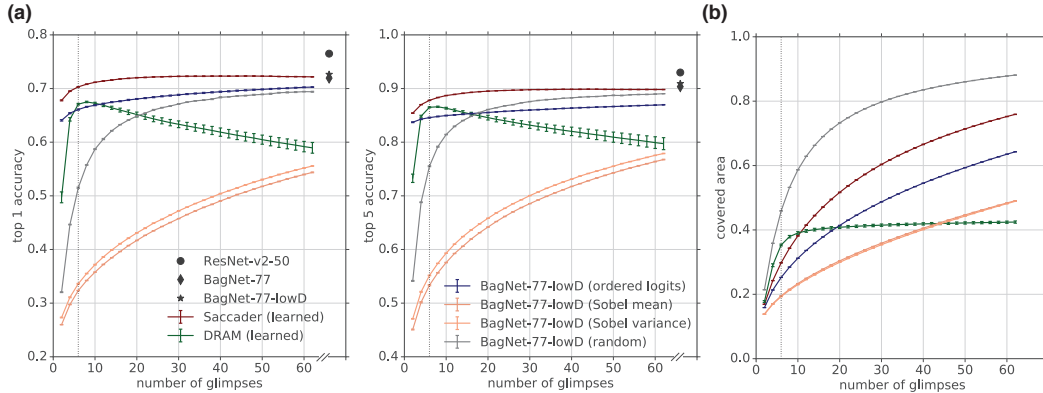

Figure 3: **Saccader makes accurate predictions with few glimpses. (a)** Top-1 and top-5 test accuracy on $224 \times 224$ images from ImageNet as a function of the number of attention glimpses used. Traces show different models and visual attention policies; vertical dotted line indicates the number of glimpses used in training. Black markers show base networks with no visual attention. **(b)** Fraction of the image area covered by the model glimpses. Error bars indicate $\pm$ SD computed from training models from 5 different random initialization.

## 4.1 Saccader Makes Accurate Predictions on ImageNet

We trained the Saccader model on the ImageNet dataset. Our results show that, with only a few glimpses covering a fraction of the image, our model achieves accuracy close to CNN models that make predictions using the whole image (see Figure 3a,b and 4a).

We compared the policy learned by the Saccader model to alternative policies/models: a *random* policy, where visual attention locations are picked uniformly from the image; an *ordered logits* policy that uses the BagNet model to pick the top $K$ locations based on the largest class logits; policies based on simple edge detection algorithms (*Sobel mean*, *Sobel variance*), which pick the top $K$ locations based on strength of edge features computed using the per-patch mean or variance of the Sobel operator [Kanopoulos et al., 1988] applied to the input image; and the deep recurrent attention model (DRAM) from Sermanet et al. [2015], Ba et al. [2014].

With small numbers of glimpses, the random policy achieves relatively poor accuracy on ImageNet (Figure 3a, 4d). With more glimpses, the accuracy slowly increases, as the policy samples more locations and covers larger parts of the image. The random policy is able to collect features from different parts from the image (Figure 4d), but many of these features are not very relevant. Edge detector-based policies (Figure 3a) also perform poorly.

The ordered logits policy starts off with accuracy much higher than a random policy, suggesting that the patches it initially picks are meaningful to classification. However, accuracy is still lower than the learned Saccader model (Figure 3a and 4b), and performance improves only slowly with additional glimpses. The ordered logits policy is able to capture some of the features relevant to classification, but it is a greedy policy that produces glimpses that cluster around a few top features (i.e., with low image coverage; Figure 4c). The learned Saccader policy on the other hand captures more diverse features (Figure 4a), leading to high accuracy with only a few glimpses.

The DRAM model also performs worse than the learned Saccader policy (Figure 3a). One major difference between the Saccader and DRAM policies is that the Saccader policy generalizes to different times, whereas accuracy of the DRAM model does not improve when allowed more glimpses than it was trained on. Sermanet et al. [2015] also reported this issue when using this model to perform fine-grained classification. In fact, increasing the number of glimpses beyond the number used for DRAM policy training leads to a drop in performance (Figure 3a) unlike the Saccader model that generalizes to greater numbers of glimpses.

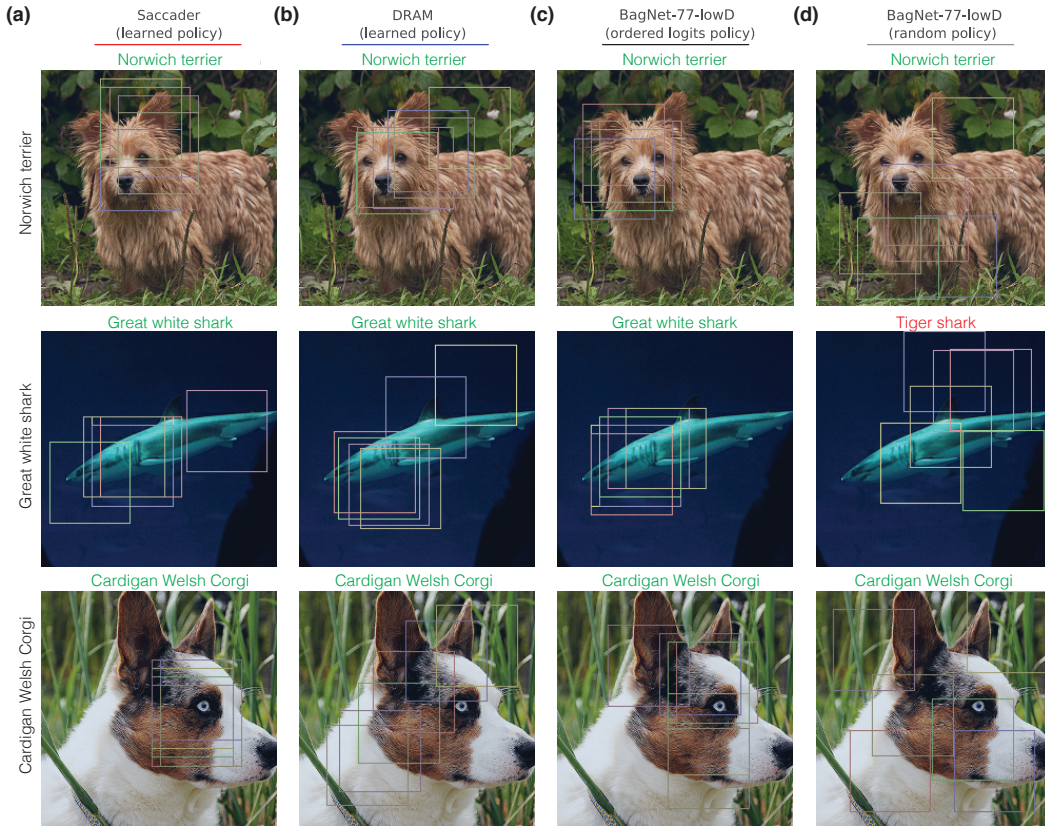

Figure 4: **Comparison of attention policies from different models.** Attention patches for different models. Green text indicates correct predictions; red text indicates incorrect predictions. **(a)** Saccader model. **(b)** DRAM model. **(c)** Policy based on the logits order from high to low across a space obtained from BagNet-77-lowD. **(d)** Random policy using BagNet-77-lowD model.

## 4.2   Saccader Attends to Locations Relevant to Classification

Glimpse locations identified by the Saccader model contain features that are highly relevant to classification. Figure 5a shows the accuracy of the model as a function of fraction of area covered by the glimpses. For the same image coverage, the Saccader model achieves the highest accuracy compared to other models. This demonstrates that the superiority of the Saccader model is not due to simple phenomena such as having larger image coverage. Our results also suggest that the pretraining procedure is necessary to achieve this performance (see Figure Supp.3 for a comparison of Saccader models with and without location pretraining). Furthermore, we show that attention network and Saccader cell are crucial components of our system. Removing the Saccader cell (i.e. using the BagNet-77-lowD ordered logits policy) yields poor results compared to the Saccader model (Figure 5a), and ablating the attention network greatly degrades performance (Figure Supp.3). The wide receptive field (RF) of the attention network allows the Saccader model to better select locations to attend to. Note that this wide RF does not impact the interpretability of the model, since the classification path RF is still limited to $77 \times 77$.

We further investigated the importance of the attended regions for classification using an analysis similar to that proposed by Zeiler and Fergus [2014]. We occluded the patches the model attends to (i.e., set the pixels to 0) and classified the resulting image using a pretrained ResNet-v2-50 model (Figures 5b and Supp.2b). Our results show that occluding patches selected by the Saccader model produces a larger drop in accuracy than occluding areas selected by other policies.

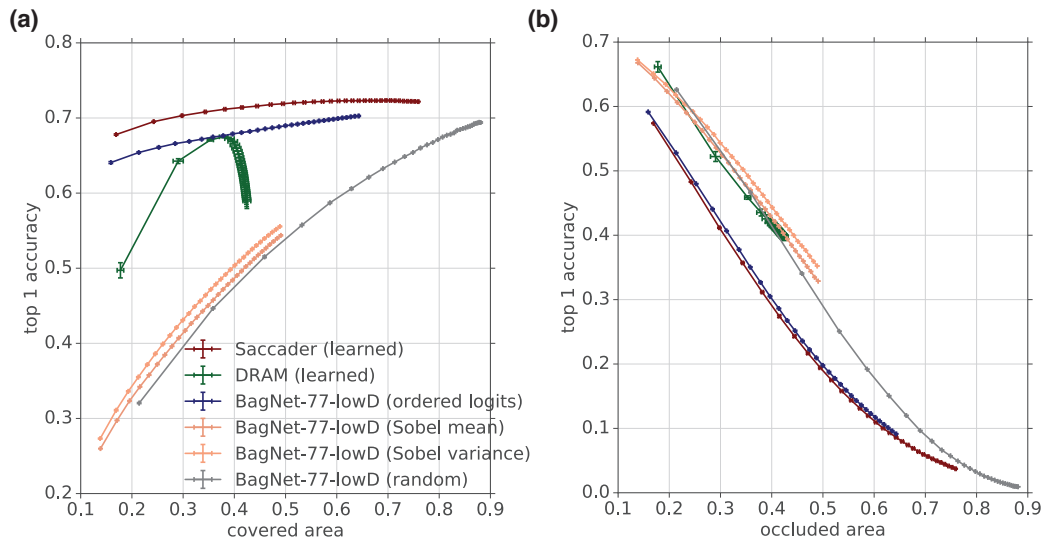

Figure 5: **Using (Occluding) Saccader glimpses gives high (poor) classification accuracy. (a)** Classification accuracy of models as a function of area covered by glimpses. The learned Saccader model achieves significantly higher accuracy while attending to a relatively small area of the image. **(b)** Classification accuracy of ResNet-v2-50 model on images where attention patches are occluded. Occluding attention patches predicted by the Saccader model significantly impairs classification. See Figure Supp.2 for top-5 accuracy. Error bars indicate ± SD computed from training models from 5 different random initializations.

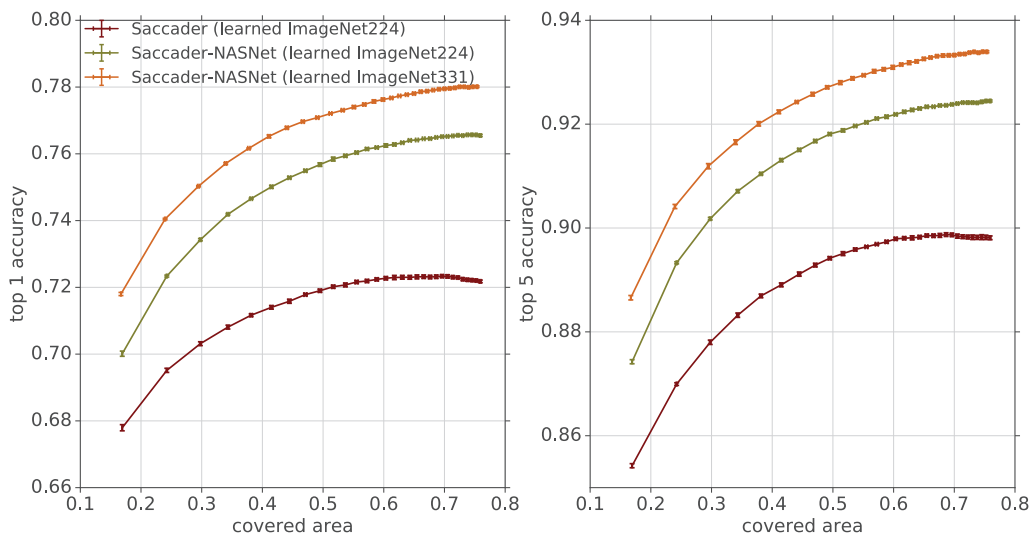

Figure 6: **Higher capacity classification network and high-resolution images further improve classification accuracy.** Graphs show top-1 and top-5 accuracy on ImageNet classification task when NASNet is used as a classification network (Saccader-NASNet) for ImageNet 224 and the higher resolution ImageNet 331. Error bars indicate ± SD computed from training models from 5 different random initializations.

## 4.3 Higher Classification Network Capacity and Better Data Quality Improve Accuracy Further

In previous sections, we used a single network to learn useful representations for both the visual attention and classification. This approach is efficient and gives reasonably good classification

accuracy. Here, we investigate if further improvements in accuracy can be attained by expanding the capacity of the classification network and using high-resolution images. In this section, we add a powerful NASNet classification network [Zoph et al., 2018] to the base Saccader model. The use of separate models for classification and localization is reminiscent of approaches used in object detection [Girshick et al., 2014, Uijlings et al., 2013, Girshick, 2015], yet here we do not have access to location labels.

We first applied the Saccader model to $224 \times 224$ pixel images to determine relevant glimpse locations. Then, we extracted the corresponding patches and used the NASNet, fine-tuned to operate on these patches, to make class predictions. Our results show that the Saccader-NASNet model is able to increase the accuracy even more while still retaining the interpretability of the predictions (Figure 6).

We investigated whether accuracy can be improved even further by training on higher resolution images. We applied the Saccader-NASNet model to patches extracted from $331 \times 331$ pixel high-resolution images from ImageNet. We down-sized these images to $224 \times 224$ and fed them to the Saccader model to identify visual attention locations. Then, we extracted the corresponding patches from the high resolution images and fed them to the NASNet model for classification (NASNet model was fine tuned on these patches). The accuracy was even higher than obtained with Saccader-NASNet model on ImageNet 224; with 6 glimpses, the top-1 and top-5 accuracy were $75.03 \pm 0.08\%$ and $91.19 \pm 0.22\%$, respectively, while processing only $29.47 \pm 0.26\%$ of the image with the NASNet.

## 5 Conclusion

In this work, we propose the Saccader model, a novel approach to image classification with hard visual attention. We design an optimization procedure that uses pretraining on an auxiliary task with only class labels and no visual attention guidance. The Saccader model is able to achieve good accuracy on ImageNet while only covering fraction of the image. The locations to which the Saccader model attends are highly relevant to the downstream classification task, and occluding them substantially reduces classification performance. Since ImageNet is a representative benchmark for natural image classification (e.g., Kornblith et al. [2019] showed that accuracy on ImageNet predicts accuracy on other natural image classification datasets), we expect the Saccader model to perform well in practical applications involving natural images. Future work is necessary to determine whether the Saccader model is applicable to non-natural image domains (e.g., in the medical field).

Although Saccader outperforms other hard attention models, it still lags behind state-of-the-art feedforward models in terms of accuracy. Our results suggest that, it may be possible to improve upon the performance achieved here by exploring larger classification model capacity and/or training on higher-quality images. Additionally, although it was previously suggested that foveation mechanisms might provide natural robustness against adversarial examples [Luo et al., 2015], the hard attention-based models that we explored here are not substantially more robust than traditional CNNs (see Appendix D). We ensure that the Saccader classification network is interpretable by limiting its input, but the attention network has access to the entire image, and thus the patch selection process remains difficult to interpret.

Here, we consider only classification task; future work can potentially extend the Saccader to many other vision tasks. The high accuracy obtained by our hard attention model and the quality of the learned visual attention policy open the door to the use of this interesting class of models in practice, particularly in applications that require understanding of classification predictions.

## Acknowledgements

We are grateful to Pieter-Jan Kindermans, Jonathon Shlens, and Jascha Sohl-Dickstein for useful discussions and helpful feedback on the manuscript. We thank Jaehoon Lee for help with computational resources.

## Footnotes

[1]Model code available at github.com/google-research/google-research/tree/master/saccader.

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
