[Supplementary Material]

# Appendix

## A Supplementary Figures

Figure Supp.1: **Examples from the Saccader model.**

Figure Supp.2: **Top-5 accuracy vs. area covered/occluded. (a)** Top-5 classification accuracy of models as a function of area covered by glimpses. The learned Saccader model achieves significantly higher accuracy while attending to a relatively small area of the image. **(b)** Top-5 classification accuracy of ResNet-v2-50 model on images where attention patches are occluded. Occluding attention areas predicted by the Saccader model significantly impairs classification. See Figure 5 for top-1 accuracy. Error bars indicate $\pm$ SD computed from training models from 5 different random initializations.

Figure Supp.3: **Pretraining and attention network helps attain high accuracy.** Comparison between Saccader models with and without location network pretraining, and with and without attention network. Error bars indicate $\pm$ SD computed from training models from 5 different random initializations.

## Representation Network

```
                    ┌─────────────────────┐
                    │   64 conv 3x3 s1    │
                    └─────────────────────┘

    ┌─────────────────────┐         ┌─────────────────────┐
    │   64 conv 1x1 s1    │         │   256 conv 1x1 s2   │
    │   64 conv 3x3 s2    │         └─────────────────────┘
    │   256 conv 1x1 s1   │
    └─────────────────────┘
              ⊕

    ┌─────────────────────┐
    │   64 conv 1x1 s1    │
    │   64 conv 3x3 s1    │
    │   256 conv 1x1 s1   │
    └─────────────────────┘
              ⊕

    ┌─────────────────────┐
    │   64 conv 1x1 s1    │
    │   64 conv 1x1 s1    │
    │   256 conv 1x1 s1   │
    └─────────────────────┘

    ┌─────────────────────┐         ┌─────────────────────┐
    │   128 conv 1x1 s1   │         │   512 conv 1x1 s2   │
    │   128 conv 3x3 s2   │         └─────────────────────┘
    │   512 conv 1x1 s1   │
    └─────────────────────┘
              ⊕

    ┌─────────────────────┐
    │   128 conv 1x1 s1   │
    │   128 conv 3x3 s1   │
    │   512 conv 1x1 s1   │
    └─────────────────────┘
              ⊕

    ┌─────────────────────┐
    │   128 conv 1x1 s1   │   x2
    │   128 conv 1x1 s1   │
    │   512 conv 1x1 s1   │
    └─────────────────────┘

    ┌─────────────────────┐         ┌─────────────────────┐
    │   256 conv 1x1 s1   │         │   1024 conv 1x1 s2  │
    │   256 conv 3x3 s2   │         └─────────────────────┘
    │   1024 conv 1x1 s1  │
    └─────────────────────┘
              ⊕

    ┌─────────────────────┐
    │   256 conv 1x1 s1   │
    │   256 conv 3x3 s1   │
    │   1024 conv 1x1 s1  │
    └─────────────────────┘
              ⊕

    ┌─────────────────────┐
    │   256 conv 1x1 s1   │   x4
    │   256 conv 1x1 s1   │
    │   1024 conv 1x1 s1  │
    └─────────────────────┘

    ┌─────────────────────┐         ┌─────────────────────┐
    │   512 conv 1x1 s1   │         │   2048 conv 1x1 s1  │
    │   512 conv 3x3 s1   │         └─────────────────────┘
    │   2048 conv 1x1 s1  │
    └─────────────────────┘
              ⊕

    ┌─────────────────────┐
    │   512 conv 1x1 s1   │
    │   512 conv 3x3 s1   │
    │   2048 conv 1x1 s1  │
    └─────────────────────┘
              ⊕

    ┌─────────────────────┐
    │   512 conv 1x1 s1   │
    │   512 conv 1x1 s1   │
    │   2048 conv 1x1 s1  │
    └─────────────────────┘

                    ┌─────────────────────┐
                    │   512 conv 1x1 s1   │
                    └─────────────────────┘
```

Figure Supp.4: **The BagNet-77-lowD architecture.** All convolutions are followed by BN+ReLU, and all $3 \times 3$ convolutions use "valid" padding. Components within the red-dashed box are used in the Saccader representation network.

Figure Supp.5: The attention network. All layers are followed by BN+ReLU.

Figure Supp.6: **The saccader cell** The component of the Saccader model that predicts the visual attention location at each time.

# B Supplementary Tables

Table Supp.1: Number of parameters.

| Model | Parameters |
|---|---|
| BagNet-77-LowD | 20,628,393 |
| ResNet-v2-50 | 25,615,849 |
| Saccader | 35,583,913 |
| DRAM | 45,610,219 |
| Saccader-NASNet | 124,537,764 |

Table Supp.2: **CNNs hyperparameters (ImageNet** $224 \times 224$**)**

| Model | learning rate | batch size | epochs | $\lambda$ |
|---|---|---|---|---|
| ResNet-v2-50 | 0.9 | 2048 | 120 | $8 \times 10^{-5}$ |
| BagNet-77 | 0.5 | 2048 | 120 | $8 \times 10^{-5}$ |
| BagNet-77-lowD | 0.5 | 2048 | 120 | $8 \times 10^{-5}$ |
| NASNet | 1.6 | 4096 | 156 | $8 \times 10^{-5}$ |
| NASNet-77* | 0.001 | 1024 | 120 | $8 \times 10^{-5}$ |

\* Fine tuning starting from a trained NASNet using crops of size $77 \times 77$ identified by Saccader.

Table Supp.3: **CNNs hyperparameters (ImageNet** $331 \times 331$**)**

| Model | learning rate | batch size | epochs | $\lambda$ |
|---|---|---|---|---|
| NASNet | 1.6 | 4096 | 156 | $8 \times 10^{-5}$ |
| NASNet-113** | 0.001 | 1024 | 120 | $8 \times 10^{-5}$ |

\* Fine tuning starting from a trained NASNet using crops of size $113 \times 113$ identified by Saccader. Note the Saccader model operates on $77 \times 77$ patches from ImageNet 224; the corresponding patches from ImageNet 331 are of size $113 \times 113$.

Table Supp.4: **DRAM hyperparameters (ImageNet** $224 \times 224$**)**

| Model | learning rate | batch size | epochs | $\lambda$ | $\nu$ |
|---|---|---|---|---|---|
| DRAM (pretraining $200 \times 200$)* | 0.8 | 2048 | 120 | $8 \times 10^{-5}$ | N/A |
| DRAM (pretraining $77 \times 77$)** | 0.001 | 2048 | 120 | $8 \times 10^{-5}$ | N/A |
| DRAM*** | 0.01 | 1024 | 120 | $8 \times 10^{-5}$ | 0 |

\* First stage of classification weights pretraining with a wide receptive field.
\*\* Second stage of classification weights pretraining with a narrow receptive field.
\*\*\* Full training starts from the model learned in the pretraining stage.

Table Supp.5: **Saccader hyperparameters (ImageNet** $(224 \times 224)$**)**

| Model | learning rate | batch size | epochs | $\lambda$ | $\nu$ |
|---|---|---|---|---|---|
| Saccader (pretraining)* | 0.2 | 1024 | 120 | N/A | 0 |
| Saccader** | 0.01 | 1024 | 120 | $8 \times 10^{-5}$ | $8 \times 10^{-5}$ |
| Saccader (no pretraining)*** | 0.01 | 1024 | 120 | $8 \times 10^{-5}$ | $8 \times 10^{-5}$ |
| Saccader (no atten. network)*** | 0.01 | 1024 | 120 | $8 \times 10^{-5}$ | $8 \times 10^{-5}$ |

\* Training for attention network weights; other weights are initialized and fixed from BagNet-77-lowD.
\*\* Weights initialized from Saccader (pretraining).
\*\*\* Training starts from BagNet-77-lowD directly without location pretraining. \*\*\*\* Modified model with attention network ablated. Representation network initialized from trained BagNet-77-lowD.

# C   Optimization and hyperparameters

Models were trained on 1,231,121 examples from ImageNet. We used 50,046 examples to tune the optimization hyper-parameters. The results were then reported on the separate test subset of 50,000 examples. We optimized all models using Stochastic Gradient Descent with Nesterov momentum of 0.9. We preprocessed images by subtracting the mean and dividing by the standard deviation of training examples. During optimization, we augmented the training data by taking a random crop within the image and then performing bicubic resizing to model's resolution. For all models except NASNet, we used a cosine learning schedule with linear warm up for 10 epochs. For NASNet, we trained with a batch size of 4096 using linear warmup to a learning rate of 1.6 over the first 10 epochs, decaying the learning rate exponentially by a factor of 0.975/epoch thereafter and taking an exponential moving average of weights with a decay of 0.9999. We used Tensor Processing Unit (TPU) accelerators in all our training.

**Convolutional neural networks**

We show the architecture of the BagNet-77-lowD model used in Saccader experiments in Figure Supp.4. For BagNet classification models, we optimized the typical cross-entropy objective:

$$J\left(\theta\right) = -\log\left(P_\theta(y_{\text{target}}|X)\right) + \frac{\lambda}{2}\sum_{i=1}^{N}\theta_i^2 \tag{7}$$

where $X$ is the input image, $y_{\text{target}}$ are the class labels, and $\theta$ are model parameters (see Table Supp.3 for hyperparameters). For the NASNet model, we additionally used label smoothing of 0.1, scheduled drop path over 250 epochs, and dropout of 0.7 on the penultimate layer.

**DRAM model**

We used the DRAM model from Ba et al. [2014] and adapted changes similar to those proposed by Sermanet et al. [2015]. In particular, the model consists of a powerful CNN (here we used ResNet-v2-50) that process a multi resolution crops concatenated along channels. The high resolution crop has the smallest receptive field, the lowest resolution crop receptive field is of the full image size, and the middle resolution crop receptive field is halfway between the highest and lowest resolution. The location information is specified by adding 2 channels with coordinate information similar to Liu et al. [2018b]. The features identified by the CNN is then sent as an input to an LSTM classification layer of size 1024. The output of the LSTM classification layer is then fed to another LSTM layer of size 1024 for location prediction. The output of the location LSTM is passed to fully connected layer of 1024 units with ReLU activation, then passed to a 2D fully connected layer with tanh activation, which represents the normalized coordinates to glimpse at in the next time step.

The best DRAM model from Sermanet et al. [2015], Ba et al. [2014] uses multi-resolution glimpses at different receptive field sizes to enhance the classification performance. This approach compromises the interpretability of the model, as the lowest-resolution glimpse covers nearly the entire image. To ensure that the DRAM model is similarly interpretable compared to the Saccader model, we limited the input to the classification component to the $77 \times 77$ high resolution glimpses and blocked the wide-receptive-field middle and low resolutions by feeding the spatial average per channel instead. On the other hand, we allowed the location prediction component of the DRAM model to have a wide receptive field (size of the whole image) by providing all three (high, middle and low) resolutions to the location LSTM layer.

Trying to enhance the accuracy of the DRAM model, we extended the pretraining of the classification weights to two stages (120 epochs on wide receptive field of size $200 \times 200$ followed by 120 epochs on the small receptive field of size $77 \times 77$. Each stage combines all the different glimpses (similar to Figure 4 Sermanet et al. [2015]) with the change of averaging the logits to compute one cross entropy loss instead of having multiple cross entropy losses for each combination of views. During pretraining, we unrolled the model for 2 time steps using a random uniform policy. We then trained all the weights with the hybrid loss specified in Ba et al. [2014].

We used REINFORCE loss weight of 0.1 and used location sampling from a Gaussian distribution with network output mean and standard deviation ($\sigma$) of 0.1 (we also tried REINFORCE loss weight

of 1. and $\sigma$ of 0.05). We also used accuracy as a baseline to center the reward and 2 MC samples to reduce the variance in the gradient estimates. We tuned the $\ell^2$-regularization weight and found that no regularization for the location weights give the best performance. Table Supp.4 summarizes the hyperparameters we used in the optimization.

**Saccader model**

Starting from a pretrained BagNet-77-LowD, we trained the location weights of the Saccader model to match the locations of the sorted logits. We then trained the whole model as discussed in Section 3.2. See Table Supp.5 for the optimization hyper-parameters.

# D   Robustness to adversarial examples

Luo et al. [2015] previously suggested that foveation-based vision mechanisms enjoy natural robustness to adversarial perturbation. This hypothesis is attractive because it provides a natural explanation for the apparent robustness of human vision to adversarial examples. However, no attention-based model we tested is meaningfully robust in a white box setting.

We generated adversarial examples using both the widely-used projected gradient descent method (PGD) [Madry et al., 2018] and the non-gradient-based simultaneous perturbation stochastic approximation (SPSA) method [Spall, 1992], suggested by Uesato et al. [2018]. We used implementations provided as part of the CleverHans library [Papernot et al., 2018]. For both attacks, we fixed the size of the perturbation to $\epsilon = 2/255$ with respect to the $\ell^\infty$ norm, which yields images that are perceptually indistinguishable from the originals. For PGD, we used a step size of $0.5/255$ and performed up to 300 iterations. Note that, although the attention mechanisms of the DRAM and Saccader models are non-differentiable, the classification network provides a gradient with respect to the input, which is used when performing PGD. For SPSA, we used the hyperparameters specified in Appendix B of Uesato et al. [2018]. Bicubic resampling can result in pixel values not in $[0, 1]$, so we clipped pixels to this range before performing attacks. We report clean accuracy after clipping.

The SPSA attack is computationally expensive. With the selected hyperparameters, the attack fails for a given example only after computing predictions for $819,200$ inputs. Thus, we restricted our analysis to 3906 randomly-chosen examples from the ImageNet validation set. We report clean accuracy after clipping and perturbed accuracy on this subset in Table Supp.6.

Table Supp.6: Adversarial robustness of models investigated to gradient-based PGD attack non-gradient-based SPSA attack, at $\epsilon = 2/255$. Results are reported on a 3906-image subset of the ImageNet validation set.

| Model | Clean Acc. | SPSA Acc. | PGD Acc. |
|---|---|---|---|
| Saccader | 70.1% | 0.3% | 0.2% |
| DRAM | 66.6% | 5.2% | 0.9% |
| ResNet-v2-50 | 76.7% | 0.1% | 0.0% |
| BagNet-77-lowD | 72.0% | 0.6% | 5.5% |

For all models we investigated, one or both attacks reduced accuracy to <1% at $\epsilon = 2/255$. Thus, at least when used in isolation, hard attention does not result in meaningful adversarial robustness. For comparison with approaches that explore robustness through training instead of model class or architecture, the state-of-the-art defense proposed by Xie et al. [2019] achieves 42.6% accuracy with a perturbation of $\epsilon = 16$, and their adversarial training baseline achieves 39.2%.

It is possible that additional robustness could be achieved by using a stochastic rather than deterministic attention mechanism, as implied by Luo et al. [2015]. However, Luo et al. [2015] only tested the transferability of an adversarial example designed for one crop to other crops, rather than constructing adversarial examples that fool models across many crops. Athalye et al. [2018] show that it is possible to construct adversarial examples for ordinary feed-forward CNNs that are robust to a wide variety of image transformations, including rescaling, rotation, translation. Moreover, non-differentiability of the attention mechanism evidently does not in itself improve robustness, and may hinder adversarial training.