[Reviews · NeurIPS 2019]

Reviewer 1



This paper addresses the problem of training hard-attention mechanisms on image classification. To do so, it introduces a new hard-attention layer (called a Saccader cell) with a pretraining procedure that improves performance. More importantly, they show that the approch is more interpretable requiring fewer glimpses than other methods while outperforming other similar approches and being close in performance to non-intepretable models such as ResNet. Originality: The proposed Saccader model is original and compares favorably to state of the art works in term of performance and also, more importantly, interpretability. Related work has been cited adequately. However, it is not clear from the paper what are the main technical difference(s) between Saccader and its main competitor DRAM. Quality: Experimental results show how the Saccader model outperforms comparable and state-of-the-art models. Indeed, figures allow us to see the differences in accuracy and in image coverage. The latter is quite informative for the interpretability claims of the paper. However, no weaknesses of the work have been noted. Indeed, while the results are important, no ablation study has been made with the Saccader cell and the attention network. These two components contain several sub-components tied together and it is not clear that they are all necessary. Furthermore, it is not clear why the attention network is needed at all. Could the increase of parameters of the attention network result in the increase performance of the Saccader model in comparison to the DRAM model? Clarity: The paper was well organized. Sections follow the usual order of NIPS papers. Small comments: In Section 3.1, item 2., it is not clear why is the “what” and “where” features are called this way. In Section 3.1, item 3., at that point in the paper, it is not clear why there is the concept of time $t$. One or two sentences grossly explaining the reinforcement learning part of the paper at that point might make the paper clearer. Significance: In addition to what has been noted in the Contributions section, while the Saccader cell and its pretraining procedure have been designed for convolutional networks, there is a safe bet that this cell can and will be used beyond computer vision tasks such as NLP and few-short learning.

Reviewer 2



This paper proposes a hard attention model named Saccader together with a pretraining procedure for efficient training of the model. The network is pre-trained in two parts using self-supervision, and the whole network is trained after that. The use of hard attention allows for understanding the image with only a small portion of the original image, and also enables understanding which part of the image is useful for classification. The authors experiment on ImageNet and discuss possible applications to other image-based tasks. Obtaining a good hard attention models is intriguing due to the computational cost it might save and the interpretability it brings. I think training a hard attention model is an interesting and important task, and the proposed model and pretraining procedures are straight-forward and seem to be well-motivated. I do have a few concerns (listed below) but at my current understanding I believe the authors have proposed an effective model that is widely applicable. Some concerns: - The experiment section could be improved. It would be helpful to include a comparison of computational cost and/or parameter counts of the Saccader to other image classification networks, either with hard attention or not. The reported accuracy could be more impressive if the size of the network is taken into account. It is probably also helpful to try the network on large-scale and/or fine-grained datasets. - I feel the description of the model could be improved in terms of clarity. For example, I don't think I see in section 3.1 how the final prediction is made--is it based purely on the "logits" at the predicted location or does the network also see the original image at given location? Also the description of the Saccader cell seems a bit fast to me, a sentence or two on what equations (1), (2), (3) do might help. - Does the "location network" in line 137 refer to the attention network, the 1-by-1 conv and the Saccader cell? Might be helpful to include an introduction before the term appears.

Reviewer 3



Strength: The idea of using hard attention for interpretability is novel to the field. Moreover, the design of the representation network limits the receptive field and prevents the model from using global information towards classification, which serves the purpose of interoperability evaluation. (However, this is also a limit which is discussed in the review) In addition, the saccader cell is also a simple yet novel design. Weakness: This model is only applicable to classification task on limited type images. Due to the design of the representation network, the model could only generate prediction on a patch of the image. The model only selects a fixed number of glimpses with fixed size for all inputs. The final prediction is a simple average across fixed number (T) of patches. This design would fail to apply on many classification tasks, such as pedestrain detection, where the image-level label is determined by multiple small ROIs. Moreover, the global distribution of spatial features is neglected by the model. The model doesn't generalize to classification tasks, such as cancer classification, where both the global features (such as the spatial distribution of radiodense tissues) and local features (lesion border) together determines the label. This model is only evaluated on ImageNet which is not representative. The pre-training procedure for the saccader cell is questionable. The loss function is designed to force the saccader cell to generate large probability on regions that representation network gives large-value logits. This step introduces a strong bias and creates a self-feedback loop. (Most of this part has been addressed by author's response.) The experiment design is somehow insufficient. First of all, the author only compares models that fit the framework of this paper (i.e. models that has an explicit glimpse selection mechanism). However, models from other families (such as weakly supervised localization models) are not compared. A simple baseline could be built using a black-box classification network (such as ResNet-V2-50) with some model-agnostic technique (such as Class Activation Map). In addition, in section 4.3, the author declares higher classification performance with NASNet but it's unclear whether this improve comes from the increased model capacity or higher input resolution.

[Author Response · NeurIPS 2019]

We thank the reviewers for their feedback. We will add the clarification sentences requested. Below are our responses:

**R1: no ablation study.** We conducted the requested study. Our results show that accuracy is poor if we ablate the attention network or the Saccader cell. We will add the analysis to Fig Supp.3 and state: "The attention network allows the Saccader to better plan for locations to attend to as it has wider receptive field (RF). Note that this wider RF does not affect the interpretability of the model as the classification path RF is still limited to 77x77. Furthermore, removing the Saccader cell (i.e. using the BagNet-77-lowD ordered logits policy) yields poor results compared to the Saccader."

**R2: computational cost and/or parameter counts.** We will add a table with the parameters count of all models. In summary, our model has 88,554,409 parameters, which is nearly twice the DRAM parameters.

**R2: how the final prediction is made.** The prediction is made using the averaged logits across the locations.

**R2: Does the "location network" ... refer to the attention network, the 1-by-1 conv and the Saccader cell?** Yes.

**R2 and R4: other datasets.** ImageNet is an extremely large and diverse dataset that contains both coarse- and fine-grained class distinctions. To achieve good performance, a method must not only distinguish among superclasses, but also e.g. among the >100 fine-grained classes of dogs. Moreover, most natural image datasets have some class overlap with ImageNet; few datasets are entirely disjoint. Recent work has suggested that ImageNet is a representative benchmark; Kornblith et al. CVPR 2019 (https://arxiv.org/abs/1805.08974) showed that accuracy on ImageNet predicts accuracy on other natural image classification datasets.

**R4: This design would fail to apply on ... pedestrain detection... cancer classification.** We agree that it would be nice to apply our method to pedestrian detection and cancer classification. However, we want to stress that natural image classification is an important computer vision problem. Recent advances in computer vision started with classification tasks on ImageNet (e.g. Krizhevsky et al., 2012) and then future research extended these methods to other domains. We will add sentences to the results to better motivate the task.

**R4: NASNet ... unclear whether this improve comes from the increased model capacity or higher input resolution.** We conducted an experiment on ImageNet 224 and using the Saccader-NASNet model. Our results show that the accuracy is better than the Saccader model alone but worse than the Saccader-NASNet model on the high resolution ImageNet 331 (we added this experiment to Figure 6). This finding demonstrates that the accuracy benefits from both the increased capacity as well as the higher input resolution.

**R4: pre-training ... This step introduces a strong bias.** We agree with the reviewer that pre-training introduces bias. However, we find that this bias is helpful in getting a better final policy. In Figure Supp3, we show that reinforcement learning after pretraining location network enhances accuracy compared to starting learning without this pretraining step. Also, note that the final learned policy is different than the pretraining target policy (i.e., the ordered logits policy). As we show in Figure 3 and 5a, the final learned policy performs much better. Just as SGD biases neural network training toward solutions that generalize well, we find that the pretraining alters the training trajectory in a way that produces a better-performing model.

**R4: baseline ... such as Class Activation Map (CAM).** In this work, we are concerned with models with hard visual attention. CAM (Zhou et al. 2016) and similar interpretability methods try to provide an explanation of the model decision in a way that relies on a heuristic (e.g., that the spatial localization of features should be preserved in the final feature map) rather than explicitly constraining how the network processes its input. These methods are fragile (see Hooker et al. 2019), and the model's final decision may nonetheless rely on information provided by features with small weights (see Jain and Wallace, 2019). Models with hard attention take a different approach by using a controller that selects parts of the input to be processed by the network, which provides interpretability by design. In our work, the representation network may be regarded as a network to construct CAM, with a guarantee that the receptive field is limited. We will add a citation to Zhou et al. 2016.

**R1: no weaknesses ... have been noted.** We will add: "Although Saccader outperforms other hard attention models, it still lags behind state-of-the-art feedforward models in terms of accuracy. Future research may extend the Saccader model to achieve even better classification performance while maintaining the interpretability of model decisions."

**R1: "what" and "where".** We will add: "These are analogous to the ventral ("what") and dorsal ("where") pathways that are involved in object recognition and localization, respectively in human vision (Goodale and Milner 1992)."

**Other improvements.** We computed errorbars (mean $\pm$ SD) for all plots. In the DRAM, we limited the high resolution to classification and the (high, mid and low) resolutions to initialize the location LSTM state, which encouraged better location exploration. We also extended the DRAM pretraining to two stages on wide and limited receptive fields (120 epochs each), and doubled the LSTM layers size. Despite these changes, the Saccader was still better than the DRAM. We improved the ResNet model accuracy in Fig 4. We corrected the Sobel and Canny baselines plots (accuracy remains poor). Since Canny and Sobel results are similar, we only included the Sobel results to improve the presentation.

[Meta-Review · NeurIPS 2019]

The reviewers are not in full agreement to accept this paper, however, I think it should be accepted. Although the model in the paper is not extremely novel, some elements of its elements are and it is interesting to see that training an accurate classifier with the hard attention mechanism is feasible even for ImageNet. Having said that, I agree with some of the critical comments in the reviews (e.g. necessity of comparisons with regard to the number of parameters and computational cost) and I encourage the authors to incorporate them into the camera-ready version of the paper.